# Ecological Effects of VR-Based Cognitive Training on ADL and IADL in MCI and AD patients: A Systematic Review and Meta-Analysis

**DOI:** 10.3390/ijerph192315875

**Published:** 2022-11-29

**Authors:** Changlae Son, Jin-Hyuck Park

**Affiliations:** 1Department of ICT Convergence, The Graduate School, Soonchunhyang University, Asan-si 31538, Republic of Korea; 2Department of Occupational Therapy, Soonchunhyang University, Asan-si 31538, Republic of Korea

**Keywords:** virtual reality, instrumental activities of daily living, activities of daily living, mild cognitive impairment, Alzheimer’s disease, cognitive training, cognitive treatment

## Abstract

Declines in activities of daily living (ADL) and instrumental activities of daily living (IADL) performances due to cognitive impairments hinder mild cognitive impairment (MCI) and Alzheimer’s disease (AD) patients’ independent and safe daily lives. In order to prevent and treat this, several cognitive interventions have been implemented, but their ecological validity was not ensured due to that their contents are far from real life. Virtual reality (VR) can resemble real life with immersive stimuli, but there have been few studies confirming its ecological effects on ADL and IADL. Therefore, this study conducted a meta-analysis of VR-based cognitive training to investigate its ecological effects on ADL and IADL in MCI and AD patients. From February 2012 to February 2022, a search was conducted for articles published in PubMed, Cochrane, Science Direct, and Web of Science. Quality assessment was assessed by the PEDro scale, and the Cochrane Collaboration tool was used to assess risk of bias. Publication bias was assessed by Egger’s regression. Five studies that met inclusion criteria were included in this study. The VR-based cognitive training showed significant effects on ADL and IADL in both MCI and AD patients. When comparing effects in each group, both MCI and AD patients showed significant effects on ADL and IADL, but MCI patients showed lower effects on ADL and IADL than AD patients. The results indicated that VR-based cognitive training would be beneficial to improve ADL and IADL in MCI and AD patients, suggesting that VR-based cognitive training is ecologically valid.

## 1. Introduction

It is estimated that the prevalence of mild cognitive impairment (MCI) worldwide ranges from 15% to 20% in older adults over 60 years old [1]. Specifically, MCI is defined as a cognitive state between normal aging and dementia and is associated with an increased risk of AD [2,3]. 

A decline in memory in MCI is highly correlated with daily lives [4,5], and an overall cognitive decline in AD interferes with safe and independent daily living [6]. In order to prevent this situation, timely interventions to minimize the progression to dementia at the MCI stage are important. There are various non-pharmacological interventions for MCI and AD patients. Even though, transfer effects of non-pharmacological interventions are unclear yet, they have been found to be effective in improving targeted cognitive functions [7]. However, their clinical effects have been mainly investigated by neuropsychological assessments. Although neuropsychological assessments have been found to be closely correlated to performances in daily lives, their ecological validity is still unclear [8]. Ecological validity refers to the degree to which assessment results are generalizable to daily lives, and neuropsychological assessments used in previous studies are removed from challenges imposed by daily life demands [8]. Most of previous studies predicted the degree to which patients can perform daily activities through traditional cognitive training [9], but due to the difference between the clinical environment and the actual environment, actual performances in real life did not match predictions. To compensate for these limitations, virtual reality (VR) technology has been widely used in clinics. In this context, VR refers to a environment or situation created by artificial technology using a computer, and the created virtual environment provides spatial and temporal experiences. Recently, VR-based cognitive training have been actively studied. As result, VR-based cognitive training has been found to stimulate and stabilize patients with cognitive impairment, suggesting VR technology could be used for clinical purposes [10,11]. Considering one of the goals of rehabilitation is to generalize clinical effects of interventions [12], VR which reduces the gap between the clinical environment and the patient’s daily living environment could be used for rehabilitation purposes with its ecological validity.

Although the use of VR technology to improve cognitive function is increasing [13], there is still controversy on its ecological effects on ADL and IADL [14]. Thus, there are limitations in the verification of the ecological validity of VR-based cognitive training. Given that VR technology could ensure the ecological validity of cognitive training, its effects would be better transferred to daily life. Therefore, this study conducted a meta-analysis to investigate the ecological effects of VR-based cognitive training on ADL and IADL in MCI and AD patients.

## 2. Materials and Methods

### 2.1. Search Strategy

We searched articles from February 2012 to 4 February 2022. Four databases were searched (PubMed, Cochrane, Web of Science, Science Direct) by combining keywords. Search keywords included “VR”, “Virtual Reality”, “MCI”, “Mild Cognitive Impairment”, “AD”, “Alzheimer’s Disease”, “Ecological validity”, “IADL”, and “ADL”. This study was registered at the PROSPERO (ID: CRD42022356079).

### 2.2. Eligibility Criteria

The eligibility criteria for this study were as follows: (1)Study design—randomized controlled trials (RCTs);(2)Participants—MCI, questionable dementia, and AD;(3)Intervention—VR-based cognitive training;(4)Controls—conventional cognitive training;(5)Outcome—ADL and IADL;(6)Language—English or Korean;(7)Full text article.

### 2.3. Article Selection

The article search and selection process were reviewed through the title and abstract of searched articles after the primary database search and, in the full review, two authors (C.S. and J.-H.P.) finally selected the articles by considering the eligibility criteria. This process was performed using a preferred reporting items for systematic reviews and meta-analysis (PRISMA) flow chart [15].

### 2.4. Quality Assessment and Risk of Bias

The qualitative evaluation of the eventually selected articles was conducted by applying the PEDro scale. The PEDro scale is a tool that analyzes the internal validity of a study with 10 items, and its grade is classified from 1 to 10 [16]. Cochrane Collaboration Review Manager (RevMan, Copenhagen, Denmark) software, version 5.4, was used to assess the risk of bias. The risk of bias evaluation was evaluated as low, uncertain, and high according to the research method. To assess the quality of the articles and to assess the risk of bias, two authors independently assessed and identified disagreements through discussion.

### 2.5. Statistical Analysis

All analyses were conducted by Comprehensive Meta-Analysis 2.0 (Biostat, Englewood, NJ, USA). Statistical heterogeneity, effect size, sensitivity analysis, and publication bias were analyzed. Statistical heterogeneity was assessed by I². Heterogeneity was acceptable when I² < 50%. When I² was less than 50%, the fixed-effects model was used. Otherwise, a random-effects model was used [17]. Hedge’s g was used to calculate and interpret the effect size. For calculation and analysis of results, mean, standard deviation, and number of subjects were used as values. Pooled effect sizes and directions of the finally selected articles were visually analyzed using a forest plot. Publication bias refers to an error in which research results are published or not published depending on the characteristics or direction of research results. If a distorted sample of studies is included in a meta-analysis, the overall size of the analysis result can be said to be a distorted result [18]. To confirm this tendency, it was reviewed and presented through a funnel plot and Egger’s regression test. Sensitivity analysis is the process of showing whether a result is reliable under various conditions. There are several methods to perform sensitivity analysis but, in this study, Hedge’s g produced results through the remaining studies except for studies that showed results more than twice that of other studies. In a meta-analysis, moderator effect analysis can more directly determine the effect size differences between subgroups and the effect of variables affecting the mean effect size. A meta-analysis of variance (ANOVA) was used in this study for analysis through the moderator effect analysis.

## 3. Results

### 3.1. Article Selection

A total of 1605 articles were searched, and these confirmed the ecological validity through results of IADL and ADL. Of these, 240 articles were excluded as being duplicates, and 1289 articles were additionally excluded based on their title and abstract. Among the remaining 76 articles, 5 articles were finally selected, excluding 71 articles that did not use VR-based intervention and did not evaluate ADL or IADL (Figure 1). 

### 3.2. Description of Included Studies

#### 3.2.1. Quality Level of Articles

Five articles selected for this study were all of the RCT design, and the their quality level was ’good’ or ’ very good’, with a score of 8 or more. However, all articles did not use a double-blinded design, and only two articles were blinded to the assessors. Details were shown in Table 1.

#### 3.2.2. Risk of Bias

Four out of the five articles reported an appropriate randomization method, and one article did not random sequence generation. Allocation concealment was reported as a low risk in all articles. Four of the five articles reported unclear risk of bias, and one article reported a high risk of bias in the blinding of participants and personnel. One article was reported as having an unclear risk at blinding of outcome assessment because there was no mention of whether an assessor was blind or not. Another article was reported as high-risk at blinding of outcome assessment because of the open-label trial design being blinded to the outcome evaluation. Four articles were reported as being low-risk in incomplete outcome data, selective reporting and other bias. The risk of bias of the included trails was presented in Figure 2. 

#### 3.2.3. Characteristics of Included Articles

In 5 articles, 74 subjects were recruited in age from their late 60 s to late 80 s. The average session time was 50 minutes, and sessions were conducted twice a week for about 10 weeks. For VR-based cognitive training, four studies used tasks related to daily lives, such as shopping and kitchen activities, and there was one study using the sports games of the Nintendo Wii. The participants of four articles were MCI patients, and the other study included AD patients. In one article, no intervention was applied to the control group, whereas conventional training or a mixture of cognitive training and physical activity was applied in the other studies. There were no articles using ADL as an outcome measure. The IADL measurements have ecological validity by measuring how much independent daily life tasks, such as finance management, are performed. The characteristics of the included articles were presented in Table 2.

#### 3.2.4. Pooled Effect Size

In order to investigate the ecological validity, only IADL results were extracted from the included articles, using various evaluation tools in to calculate an effect size. 

### 3.3. Meta-Analysis Results

#### 3.3.1. Effect Size of VR-Based Cognitive Training on IADL

There was no considerable heterogeneity across the included studies (I² = 26.71, *p* = 0.243), so a fixed-effects model was used to evaluate effect size. We found that effect sizes of VR-based cognitive training were significantly greater compared to conventional cognitive training (*g* = 0.558, *p* = 0.001), suggesting that VR-based cognitive training could be more beneficial to improve IADL in MCI and AD patients (Figure 3).

#### 3.3.2. Sensitivity Analysis

In one article, Hedge’s *g* was more than twice that of the other four articles. If the results were calculated except for this article, Hedge’s *g* was 0.439, which was interpreted as the medium effect size. The effect size was significant (*p* < 0.05). It was presented in Figure 4.

#### 3.3.3. Moderator Effect Analysis

Effect sizes were analyzed by dividing subjects into MCI and AD patients. As a result of analyzing four articles concerning MCI and one article concerning AD patients, the AD group showed a high heterogeneity (I² = 79.18), but Hedge’s *g* had a medium effect size of 0.623 (*p* < 0.05). For MCI, Hedge’s *g* had a medium size of 0.523 (*p* < 0.05) with a low heterogeneity (I² = 0). 

#### 3.3.4. Publication Bias

A funnel plot was used to evaluate publication bias. Figure 5 showed that except for one article, four articles were distributed within the same area, showing a symmetrical tendency. The result of the Egger’s regression test was 1.72 (*p* = 0.16). There was no publication bias, as the results were not statistically significant.

## 4. Discussion

The purpose of this study was to investigate the ecological effects of VR-based cognitive training on ADL and IADL in MCI and AD patients. As the results indicate, VR-based cognitive training showed significant improvements in IADL in MCI and AD patients, with medium effect sizes. This result was consistent with a recent meta-analysis reporting small sizes for VR-based cognitive training on IADL [13]. However, while a previous meta-analysis mainly investigated the effects of VR-based cognitive training on cognitive domain and reported small effect sizes on IADL, this study focused on its ecological validity and confirmed medium effect sizes on it. This could be attributed to differences in samples and the quality of included studies between a previous meta-analysis and this study.

In a previous study, VR-based cognitive training was shown to be effective in improving IADL in patients with MCI [20]. Similarly, in another previous study with AD patients, VR-based cognitive training was found to have a clinical effect [19], supporting our results. However, in contrast to previous studies, this meta-analysis included articles of high quality, which may have affected our findings. In other words, when conclusions were drawn based on high quality studies, it was also confirmed that VR-based cognitive training is helpful in improving IADL in both MCI and AD patients, suggesting that VR-based cognitive training is ecologically valid.

Ecological effects of VR-based cognitive training on IADL in patients with MCI and AD are attributed to the characteristics of VR. Virtual reality can replace an individual’s external environment and senses with an artificial environment that updates according to the individual’s orientation and body movement [23]. Furthermore, VR simulates an immersive and interactive environment, providing a feeling of ‘being there’. Thus, assessments or interventions performed in VR environments could be tailored to the needs of subjects [10]. These characteristics of VR could be a key factor for ensuring its ecological validity [24].

The original purpose of this study was to investigate the ecological effects of VR-based cognitive training on ADL and IADL. However, none of the included studies in this meta-analysis evaluated ADL, meaning that VR’s ecological effects on ADL were not measured. In a previous study, IADL assessments were more closely related to cognitive functions than ADL assessments. Therefore, to measure the transfer effects of cognitive function training, IADL was mainly assessed as outcome measures [25], which is consistent with this study. Indeed, ADL consists of activities which are necessary for self-care, such as bathing and dressing, while IADL involves requiring more complex and cognitive functions, such as cooking, financial management, transportation, and shopping [26,27].

A recent meta-analysis did not evaluate both MCI and AD patients, but only MCI patients or only AD patients [13]. Considering that previous meta-analyses investigated the ecological effects of VR-based cognitive training in MCI or AD patients rather than including both groups, the positive findings of this study including both groups have originality [13,28]. This study confirmed the significant medium effects of VR-based cognitive training on IADL not only grouping MCI and AD patients together, but also by dividing both groups. Specifically, effect sizes in MCI patients were lower than those in AD patients, which is contrary to our expectation that MCI patients would show a greater benefit from VR-based cognitive training in comparison to AD patients. Since patients with MCI show less cognitive impairment than those with AD, they tend to show a higher therapeutic effect [28,29]. However, in the case of AD, since the number of articles was only one, there was a limit to interpreting the results by dividing them into two groups.

Even though this study has clinical implications, there were several limitations. Firstly, the number of selected articles was small. Specifically, only one had confirmed the effects of VR-based cognitive training on IADL in AD patients. Furthermore, the small number of subject in the included studies could bring the bias in estimated effect sizes. Thus, the findings of this meta-analysis need to be interpreted with caution. Secondly, there was no study evaluating the long-term effects of VR-based cognitive training. Thirdly, the selected articles were all not double-blinded, resulting in a potential for bias. 

In this study, ecological effects of VR-based cognitive training were confirmed in patients with MCI and AD. The current finding suggested that the use of VR could increase the ecological validity of cognitive training and that it could be implemented in safe ways. However, a previous study on the usability issues of VR systems with older adults found it hard for them to learn to use controllers because of their memory impairment and unfamiliarity with them. Therefore, to address this issue, it is recommended to use VR systems when users become accustomed to using controllers through a sufficient period of practice. On the other hand, the characteristics of VR-based cognitive training vary across the included studies, which could be a limitation to adapting VR as a treatment tool in a clinical setting. Nevertheless, VR systems could ensure ecological validity by providing an environment which is as similar to daily lives as possible, and this nature of VR brings clinical implications. Due to the small number of articles included in this meta-analysis, this study shed new light on this clinical implication rather than statistical values, such as effect sizes. 

## 5. Conclusions

We found that VR-based cognitive training is effective in improving IADL in patients with MCI and AD. This is because VR-based cognitive training could resemble real-life environments and daily life tasks similar to traditional interventions. The difference between clinical and actual daily life could be reduced through its higher ecological validity than conventional intervention. This finding sheds new light on the potential use of VR as an alternative option for cognitive training to ensure its ecological validity. We hope that the current findings will provide a guide that cognitive training needs to be implemented in an environment centered on daily life tasks including cognitive components in order to transfer the effects of cognitive training to everyday life.

## Figures and Tables

**Figure 1 ijerph-19-15875-f001:**
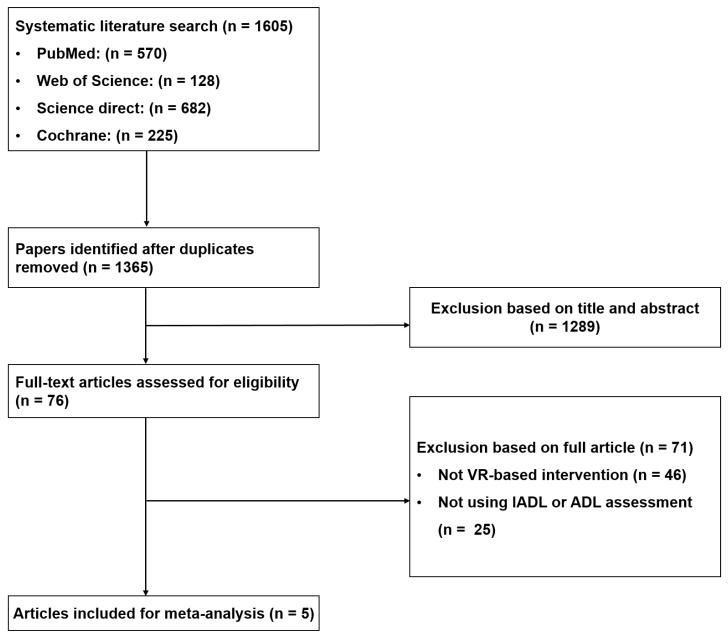
Article screening flow chart.

**Figure 2 ijerph-19-15875-f002:**
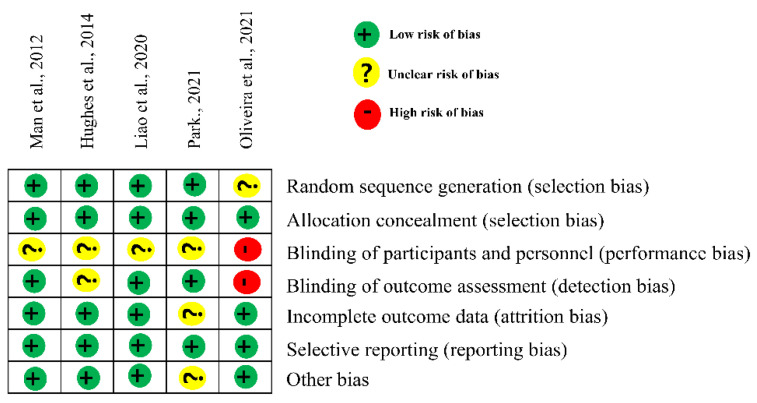
Risk of bias summary. Park., (2021) [14], Oliveira et al., (2021) [19], Liao et al., (2020) [20], Hughes et al., (2014) [21], Man et al., (2012) [22].

**Figure 3 ijerph-19-15875-f003:**
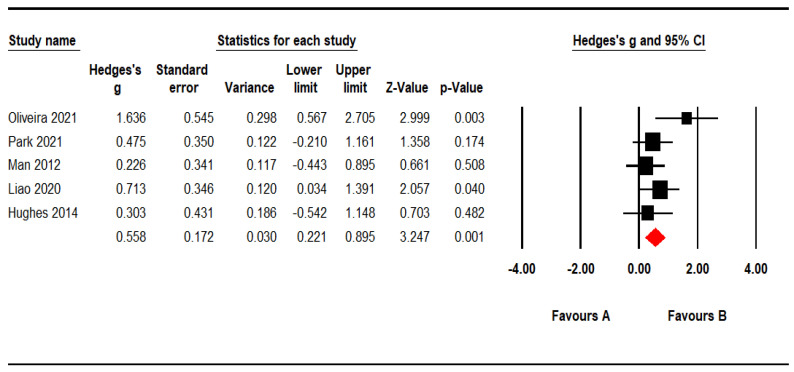
Forest plot demonstrating the efficacy of VR-based cognitive training on IADL in MCI and AD patients. Park., (2021) [14], Oliveira et al., (2021) [19], Liao et al., (2020) [20], Hughes et al., (2014) [21], Man et al., (2012) [22].

**Figure 4 ijerph-19-15875-f004:**
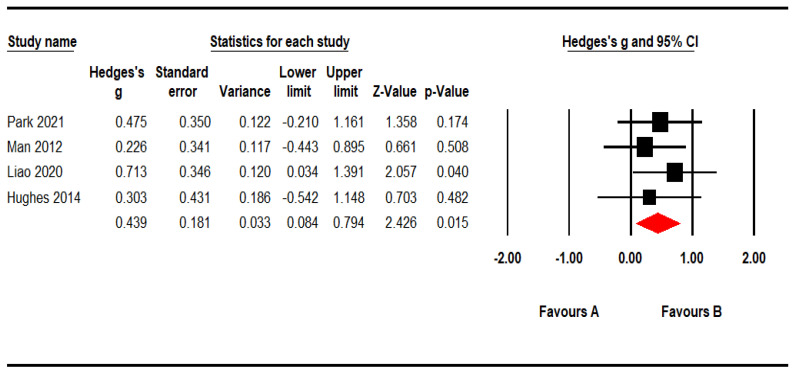
Sensitivity analysis of the IADL effect of using VR-based cognitive intervention on MCI and AD patients (I²= 0). Park., (2021) [14], Oliveira et al., (2021) [19], Liao et al., (2020) [20], Hughes et al., (2014) [21], Man et al., (2012) [22].

**Figure 5 ijerph-19-15875-f005:**
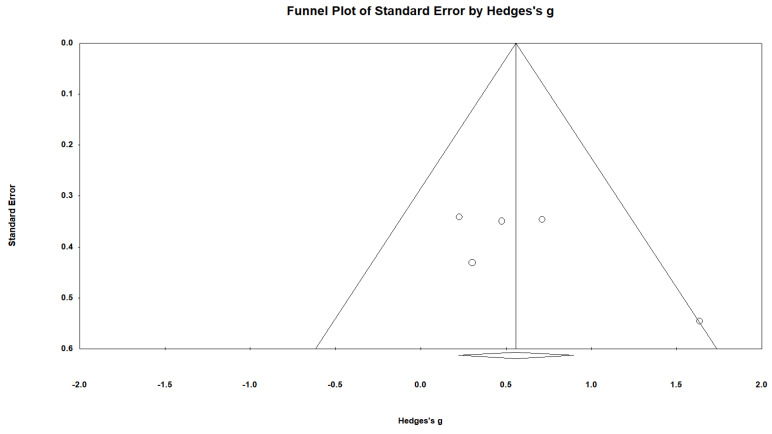
Publication bias of the included articles.

**Table 1 ijerph-19-15875-t001:** PEDro scale score of included articles.

No.	Quality Assessment Scale	Man et al., (2012) [22]	Hughes et al., (2014) [21]	Liao et al., (2020) [20]	Park., (2021) [14]	Oliveira et al., (2021) [19]
1	Eligibility criteria were specified.	Yes	Yes	Yes	Yes	Yes
2	Subjects were randomly allocated to groups.	Yes	Yes	Yes	Yes	Yes
3	Allocation was concealed.	Yes	Yes	Yes	Yes	Yes
4	The groups were similar at baseline regarding the most important prognostic indicators.	Yes	Yes	Yes	Yes	Yes
5	There was blinding of all subjects.	No	No	No	No	No
6	There was blinding of all therapists who administered the therapy.	No	No	No	No	No
7	There was blinding of all assessors who measured at least one key outcome.	No	No	Yes	Yes	No
8	Measures of at least one key were obtained from more than 85% of the subjects initially allocated to groups.	Yes	Yes	Yes	Yes	Yes
9	Intention to treat.	Yes	Yes	Yes	No	Yes
10	The results of between-group statistical comparisons are reported for at least one key outcome.	Yes	Yes	Yes	Yes	Yes
11	The study provides both point measures and measures of variability for at least one key outcome.	Yes	Yes	Yes	Yes	Yes
Sum score		8	8	9	8	8

**Table 2 ijerph-19-15875-t002:** Characteristics of the included articles.

	Study		Characteristics of Participants			Intervention		Outcome
	Author	Study Design	Numbers	Age (SD)	Session	EG	CG	
1	Oliveira et al., (2021) [19]	RCT	EG: 10 CG: 7	83.24 (5.66)	45 min, 2 times/week, 9 h	Systemic Lisbon Battery (IADL tasks such morning hygiene and kitchen)	Usual care units for older adults	The Lawton–Brody IADL scale
2	Park., (2021) [14]	RCT	EG: 16 CG: 16	EG: 72.25 (5.13) CG: 70.88 (4.51)	2 times/week, 8 weeks	Virtual supermarket shopping	X	K-IADL
3	Liao et al., (2020) [20]	RCT	EG: 18 CG: 16	EG: 75.50 (5.20) CG: 73.10 (6.80)	60 min, 3 times/week, 12 weeks	Mass rapid transit/looking for a store/kitchen chef/convenience store clerk	Combined physical and cognitive training	The Lawton IADL Scale
4	Hughes et al., (2014) [21]	RCT	EG: 10 CG: 10	EG: 78.50 (7.10) CG: 76.20 (4.30)	90 min, 1 h/week, 24 weeks	Nintendo Wii sports game	Healthy aging education program	The Timed IADL
5	Man et al., (2012) [22]	RCT	EG: 20 CG: 24	EG: 80.30 (1.21) CG: 80.28 (1.31)	30 min 2–3 times/week, 10 sessions	Home setting and convenience shop management	Therapist-led training	HK Lawton IADL

Abbreviations are as follows: RCT = randomized controlled trial; EG = experimental group; CG = control group; IADL = instrumental activities of daily living.

## Data Availability

Not applicable.

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
