# Peer review of "Ecological Effects of VR-Based Cognitive Training on ADL and IADL in MCI and AD patients: A Systematic Review and Meta-Analysis"

_ijerph, 2022, doi:10.3390/ijerph192315875_

Round 1
Reviewer 1 Report
Virtual reality-based brain cognitive therapy or rehabilitation medical intervention for MCI patients must be a very important topic, and the reason may be due to the past 30 years of clinical use and countless papers. In that context, this paper needs to be highly evaluated for its timeliness. This is because synthesizing various papers and analyzing them in the quantitative way as much as possible is an essential academic process; It is a method of collecting all individual data related to a particular research problem in one place, effectively combining individual research results into a large single study, and then analyzing it statistically.
The choice of effect size may vary depending on the research design, result measurement method, and statistical application method used in individual studies. Most of the effect sizes used in the study can be classified into one of D family (standardized mean difference, etc.), Odds ratio family (probability ratio, etc.), and r family (correlation coefficient, etc.). I hope that this part will be described in more detail in this paper.
In general, meta-analysis is performed in the order of calculating the summary estimate of individual studies first, weighting each study, and then calculating the combined summary statistics of each study using the weighted average, which would be nice to explain in more detail in this paper.
Author Response
Thank you for your comment. We have revised our manuscript to reflect your comment as much as possible. Please refer to the attached file.

Reviewer 2 Report
Review meta analysis VR training
I have read this paper with interest. It addresses an important topic – how VR training can improve performance of patients in daily life. I think the meta-analysis has been conducted carefully. I do have a number of major and minor concerns though.
Major concerns
Ø Please clarify your expectations regarding the use of VR technology. It is now unclear whether it could be useful because it allows more ecologically valid assessments of performance levels that generalize to performance in the real world (i.e. better diagnostics) or whether training in VR causes better real world performance (i.e. better interventions). This should be clearer in the paper and in its motivation.
Ø Not only the number of studies included in the paper is small (5) but also the number of participants per study. Please consider including a small samples / small study effects meta-analysis (Nuijten et al., 2015).
Ø Please explain how the present study differs from Zhong, D.; Chen, L.; Feng, Y.; Huang, L.; Liu, J.; Zhang, L. Effects of virtual reality cognitive training in individuals with mild 303 cognitive impairment: A systematic review and meta-analysis. Int. J. Geriatr. Phychiatry. 2021, 36, 1829-1847. [Crossref]
Is it because you now include other patient groups as well (AD). What is the reason to do this follow up?
Ø Did you expect any differences for the AD and the MCI groups? Please discuss this at more length in the Discussion.
Ø Discuss further what the nature of the VR training was in the various studies and why they could have their rehabilitation potential. A limitation could be that the type of training differed between studies and was not focused/ elaborate enough.
Minor points
Ø Please check the English language. Document contains several typos and inappropriate formulations.
Ø I agree that the ecological validity of neuropsychological assessments in clinical settings can be doubted but now this is formulated too strongly.
Ø I recommend to elaborate a bit further on how clinical assessments relate to daily life performance, and how VR can help to assess cognitive performance and can be used to improve performance in specific cognitive domains. There are more studies to be discussed (see Spreij, L. A., Visser-Meily, J. M. A., Van Heugten, C. M., & Nijboer, T. C. W. (2014). Novel insights into the rehabilitation of memory post acquired brain injury: A systematic review. Frontiers in Human Neuroscience, 8, 1-19.; Spreij L.A., Visser-Meily J.M.A., Sibbel J., Gosselt I.K., Nijboer T.C.W. Feasibility and user-experience of virtual reality in neuropsychological assessment following stroke (2020).
Ø The claims about the effectiveness of non-pharmacological interventions are formulated too strongly. Discuss more references and the degree to which performance can be improved.
Ø What is the ecological validity of the ADL measures?
Author Response

(The authors gave the same response as above.)

Reviewer 3 Report
The aim of this paper was to examine Virtual Reality (VR) ecological effects on Activities of Daily Living (ADL) and Instrumental Activities of Daily Living (IADL) for people with Mild Cognitive Impairment (MCI) and Alzheimer’s Disease (AD). This manuscript is well written and easy to read and followed the PRISMA guidelines. The study was registered at PROSPERO.
I consider “Introduction” section interesting, and the references are sufficient and up to date; however, if you want to publish in an international journal why do you focus on data from the Korean Dementia Observatory?
Regarding “Methods” section, why did you choose Korean as language? The VR programs included in the study are quite different, e.g. Systemic Lisbon Battery vs Nintendo Wii or others. Perhaps it would be interesting to describe them a little better. Also, I think the flow chart could be more rigorous (for example, indicating how many articles were selected considering the sources).
In “Discussion”, precautions, and risks of the use of VR for this population should be discussed in greater depth. The fact that this article is being published in a special theme (Virtual Reality-Based Cognitive Training for Cognitive Function and Psychological Symptoms) I expected the authors to have a richer discussion. Due to Alzheimer’s Disease progressive nature, symptoms can vary from mild memory loss to complete lack of ability to respond to one’s surroundings. The memory impairments brought on by this disease can lead to specific problems with memory interference, which may be caused by dysfunction in working and semantic memory. When conducting experiments with VR on Alzheimer’s patients, there is also the added difficulty of the individual having trouble remembering the instructions and needing external cues to complete memory tasks.
In “Conclusion” the authors mention that they want this article to help design cognitive training programs, for that they have to give clearer guidelines on what to do in the “Discussion” section. Maybe the authors could discuss some effective design criteria and strategies and suggest some possibly useful protocols and procedures.
I recommend that the clinical and social implications of this study emerge in a more pragmatic way. Caution should be taken when interpreting this study’s results given the significant limitations namely only including 5 studies with small samples.
Further Reading:
https://alz-journals.onlinelibrary.wiley.com/doi/10.1002/alz.049301 --> A systematic review on the effectiveness of virtual reality cognitive training (VRCT) and computer-based cognitive training (CBCT) for individuals with mild cognitive impairment (MCI)
https://www.frontiersin.org/articles/10.3389/fdgth.2022.916052/full --> Enhance VR: A Multisensory Approach to Cognitive Training and Monitoring
Author Response

(The authors gave the same response as above.)

Round 2
Reviewer 3 Report
In my perspective, the authors made the necessary changes in order to answer to my comments and suggestions. I consider that the article as it is can be published.